# In vitro *Edwardsiella piscicida* CK108 Transcriptome Profiles with Subinhibitory Concentrations of Phenol and Formalin Reveal New Insights into Bacterial Pathogenesis Mechanisms

**DOI:** 10.3390/microorganisms8071068

**Published:** 2020-07-17

**Authors:** Ju Bin Yoon, Sungmin Hwang, Se-Won Baek, Seungki Lee, Woo Young Bang, Ki Hwan Moon

**Affiliations:** 1Division of Marine Bioscience, Korea Maritime and Ocean University, Busan 49112, Korea; fishbin12345@gmail.com (J.B.Y.); bsw8888@kmou.ac.kr (S.-W.B.); 2Department of Biology, Duke University, Durham, NC 27707, USA; sungmin.hwang@duke.edu; 3National Institute of Biological Resources, Environmental Research Complex, Incheon 22689, Korea; metany@korea.kr

**Keywords:** *Edwardsiella piscicida*, fish pathogen, sub-inhibitory concentration, formalin, phenol, differentially expressed genes, motility, virulence factors

## Abstract

Phenol and formalin are major water pollutants that are frequently discharged into the aquatic milieu. These chemicals can affect broad domains of life, including microorganisms. Aquatic pollutants, unlike terrestrial pollutants, are easily diluted in water environments and exist at a sub-inhibitory concentration (sub-IC), thus not directly inhibiting bacterial growth. However, they can modulate gene expression profiles. The sub-IC values of phenol and formalin were measured by minimal inhibitory concentration (MIC) assay to be 0.146% (1.3 mM) and 0.0039% (0.38 mM), respectively, in *Edwardsiella piscicida* CK108, a Gram-negative fish pathogen. We investigated the differentially expressed genes (DEG) by RNA-seq when the cells were exposed to the sub-ICs of phenol and formalin. DEG analyses revealed that genes involved in major virulence factors (type I fimbriae, flagella, type III and type VI secretion system) and various cellular pathways (energy production, amino acid synthesis, carbohydrate metabolism and two-component regulatory systems) were up- or downregulated by both chemicals. The genome-wide gene expression data corresponded to the results of a quantitative reverse complementary-PCR and motility assay. This study not only provides insight into how a representative fish pathogen, *E. piscicida* CK108, responds to the sub-ICs of phenol and formalin but also shows the importance of controlling chemical pollutants in aquatic environments.

## 1. Introduction

Environmental pollutants are gradually increasing, along with the rising demands of chemical products and energy generation due to population growth and industrial development. An enormous amount of chemical usage and inappropriate chemical controls threaten many ground-based ecosystems. Moreover, a high concentration of terrestrial contaminants flowing into water environments can be fatal to microorganisms that thrive in aquatic habitats [1,2,3]. However, pollutants can be rapidly diluted to a low concentration, a so-called sub-inhibitory concentration (sub-IC), after inflow into water. While the sub-IC of pollutants would not have a deadly impact on microorganisms, it could be a signal that could alter gene expression profiles.

The US Environmental Protection Agency has listed several environmental pollutants [4]. Phenolic compounds, one type of the listed chemicals, are generated by natural processes and present in limited amounts in nature, however, the human-related large volume of phenol production and leakage from petroleum refining, the petrochemical industry, pharmaceutical companies, electronics factories and coal conversion factories is fatal to living organisms including humans [5,6]. Although there has been a study that showed phenol exposure inhibits biofilm formation in *Nitrosomonas europaea*, the comprehensive toxic mechanism of phenol to marine microorganisms is poorly understood [4,7,8]. In addition, the frequency of drug usage tends to multiply at fish farms, as the rate of disease in aquaculture organisms increases due to the rapid growth of the aquaculture industry worldwide [9]. Formalin has been widely used to control against external parasitic infestation and to prevent bacterial contamination in the aquaculture industry [10]. Therefore, the accumulation of formalin use and drainage into the ocean from fish farms can cause marine pollution.

*Edwardsiella piscicida*, a rod-shaped Gram-negative bacterium, is known to be a critical fish pathogen that causes severe symptoms (i.e., septicemia and organ necrosis) and disease (i.e., edwardsiellosis) in fish such as flounder and catfish. Fish lethality derived from the virulence factors of *E. piscicida* causes enormous economic losses so various types of antibiotics and chemicals are utilized in the aquaculture industry [11,12]. The pathogenesis of *E. piscicida* appears to be multifactorial virulence factors such as a type III secretion system (T3SS), type VI secretion system (T6SS), flagellin and a two-component system [13]. T3SS and T6SS are major virulence factors that play important roles in the adherence, penetration, survival and replication of *E. piscicida* in epithelial cells and phagocytes [11]. Flagellins are motility-related proteins that are important for penetration into the epithelial cells of hosts [11]. Two-component systems such as EsrA-EsrB, PhoP-PhoQ and QseB-QseC also act as virulence regulators [14].

Understanding how bacteria cope with inconsistent changes in natural environments is one of the critical research subjects in microbiology. Transcription is an initial process that enables microorganisms to respond to the fluctuating environmental conditions. Various experimental tools have been extensively developed, not only to measure the expression of a few target genes by quantitative reverse transcription PCR and Northern blotting [15,16] but also to quantify the global gene expression across the whole genome with microarray- and next-generation sequencing-based RNA-seq from all domains of life [17,18,19]. Here, we used genome-wide transcriptional profiling with RNA-seq to understand the metabolic alterations when *E. piscicida* CK108 is exposed to the sub-ICs of an aquatic pollutant and a disinfectant widely used in the fish industry—phenol and formalin, respectively. This work provides not only the fundamental transcriptomic knowledge of the marine pathogen when phenol and formalin are present but also insights of the appropriate use of chemicals to prevent from potential side effects in marine ecosystems.

## 2. Materials and Methods

### 2.1. Strain and Growth Conditions

*E. piscicida* CK108 strain was grown at 27 °C in Brain Heart Infusion (BHI) (Difco Inc., Sparks, NV, USA) medium [20]. Chemicals were added when necessary with the following sub-ICs: 0.146% (1.3 mM) phenol solution (Sigma-Aldrich, St. Louis, MO, USA) and 0.0039% (0.38 mM) formalin (Duksan, Seoul, Republic of Korea). The growth of *E. piscicida* CK108 on BHI medium was assayed at 27 °C with continuous double orbital shaking using an Epoch2 Microplate Spectrophotometer (BioTek, Winooski, VT, USA). The optical density (OD_600 nm_) was measured at 1 h intervals for the length of the experiment. For each experiment, three monoclonal cultures from each strain were included, and each monoclonal culture was used to inoculate three test wells from a 96-well microplate.

### 2.2. RNA Extraction, Library Construction, and Sequencing

To extract the total RNA, three or four single colonies of *E. piscicida* CK108 were pre-cultured in BHI medium and transferred to the same liquid to incubate at 27 °C with shaking (200 rpm) for 24 h. Phenol solution [final concentration, 0.146%, (1.3 mM)] or formalin [final concentration, 0.0039%, (0.38 mM)] were added when necessary. Total RNA was extracted with the AccuPrep Universal RNA Extraction Kit (Bioneer, Daejeon, Republic of Korea), in accordance with the manufacturer’s instructions. DNA remaining in the RNA samples was removed by the Turbo DNA-free kit (Invitrogen, Carlsbad, CA, USA) in accordance with the manufacturer’s instructions. The purity and integrity of RNAs were assessed by measuring the absorbance at 260/280 nm using an Epoch2 Microplate Spectrophotometer (BioTek, Winooski, VT, USA) and Bioanalyzer 2100 (Aligent Technologies, CA, USA). Ribosomal RNA from the purified RNA (RIN > 8) was removed by the Anydeplete probe (Nugen, Redwood City, CA, USA), and the remaining RNA was used to construct a sequencing library with the Truseq Stranded Total RNA H/M/R Prep kit (Illumina, San Diego, CA, USA) according to the manufacturer’s recommendations. The quality of the amplified libraries was verified by Bioanalyzer (Agilent, Santa Clara, CA, USA). The NovaSeq 6000 (Illumina, San Diego, CA, USA) platform was used for the sequencing, and 100-bp of pair-end reads was generated (20–30 million reads per sample).

### 2.3. RNA-seq Data Analysis

The sequencing artifacts and contamination of raw sequencing reads were assessed by FastQC v.0.11.7 (available online: https://www.bioinformatics.babraham.ac.uk/projects/fastqc) and further adapter, poor-quality or short reads (< 10 bp) were trimmed by Cutadapt v.1.18 [21]. The preprocessed reads were aligned to the reference genome of *E. piscicida* C07–087 by Bowtie2 v.2.3.4.3 [22], followed by sorting and indexing by Samtools v.1.9 [23]. The number of reads mapped to each transcript was counted by HTSeq v.0.11.0 [24]. DESeq2 v.1.26.0 was used to normalize the read counts and identify differentially expressed genes (DEGs) [25]. Genes with a log2 fold change larger than 2 and a false discovery rate (FDR) determined by Benjamini–Hochberg (BH) correction for multiple hypothesis testing of less than 0.01 were considered to be DEGs. EnhancedVolcano v.1.4.0 [26] was used for data visualization. Functional enrichment of DEGs was performed using the hypergeometric test with BH correction. Annotations were computed using the eggNOG-mapper with eggNOG 4.5 orthology data [27]. Pathway enrichment of DEGs was carried out by Pathview v.1.28.0 [28].

### 2.4. Quantitative Reverse Transcription PCR

Total RNA was isolated by an RNA extraction kit (Bioneer, Daejeon, Republic of Korea) in accordance with the procedures provided in the kit. An additional DNA digestion step was applied to ensure DNA was free from the extracted RNA using a Turbo DNA-free kit (Invitrogen, Carlsbad, CA, USA). The DNA-free RNA was confirmed by end-point PCR with 35 cycles, targeting the control gene gapdh. 2X OneStep qRT-PCR Master Mix (Biofact, Daejeon, Republic of Korea) was used to amplify the target genes using a Real-Time PCR System (Applied Biosystems, Waltham, Massachusetts, USA). Each primer set is listed in Appendix A. The specificity of the primer sets was examined by a melting curve. The expression levels of target genes were normalized to the internal gene *gapdh* (ETAC_06975), and the relative expression of target genes was calculated by Livak’s method.

### 2.5. Motility Assay

Three single colonies were grown at the early exponential phase (OD_600 nm_, ~ 0.3) in BHI broth. Five microliters of cell culture was dropped onto 0.3% (w/v) motility agar plates of BHI medium containing sub-ICs of 0.146% (1.3 mM) phenol or 0.0039% (0.38 mM) formalin. Cell motility was monitored by measuring the diameter of the swimming area on the soft agar after incubating at 27 °C for 17 h.

### 2.6. Data Availability

The raw RNA-seq data are available at the accession number, PRJNA629490.

## 3. Results

### 3.1. Determination and Growth of E. piscicida CK108 in Sub-ICs of Phenol and Formalin

We chose two chemicals, phenol and formalin, as a representative terrestrial pollutant that flows into the ocean and a disinfectant widely used in the fish industry to control hygiene, and we sought to determine how the chemicals influence a fish pathogen model organism, *E. piscicida* CK108 [20,29]. To understand the genetic and metabolic conversion at the sub-inhibitory concentration (sub-IC), we first investigated the cell growth on the various concentrations of phenol and formalin in vitro. Cells were grown when the phenol concentration was less than 0.146% (1.3 mM) (Figure 1A) or the formalin concentration was less than 0.0039% (0.38 mM) (Figure 1B). We used these chemical concentrations as the sub-ICs and monitored the cell growth. Regardless of the growth conditions, cells reached almost the same cell density in the stationary phase (Figure 2A). However, while cells with phenol showed a similar growth pattern as non-treated cells, cells in the formalin-added condition took 2-fold longer in the lag phase than in other growth conditions (Figure 2A). Moreover, compared with non-treated cell, the specific growth rate of cells with phenol and formalin was significantly decreased to 16% and 57%, respectively (Figure 2B). Altogether, the growth assay revealed that the growth of *E. piscicida* CK108 is distinctive in response to chemicals. This suggests that different cellular metabolism processes are initiated when phenol and formalin are present.

### 3.2. Overall Genome-Wide Gene Expression Profiling

Global transcriptomic profiling provides quantitative information about the whole population of microbial RNA on the specific growth conditions. To investigate the genome-wide gene expression level, we performed RNA-seq analysis using the total mRNA purified from the exponential growth phase of *E. piscicida* CK108 when exposed to phenol or formalin. The transcriptomic profiles of cells treated with phenol or formalin were compared with those from untreated cells. To assess the inter- and intra-sample differences/similarities, the gene expression levels of each biological replicate were monitored by hierarchical clustering with the Euclidean distances. The relative clustering of transcriptomes was related to the growth conditions (Appendix A), and the low variation between the biological replicates was observed from phenol- or formalin-treated samples. In addition, the same trend was seen for the principle component analysis. The variance of the first and the second principal components was 53% and 41%, respectively (Appendix A). These results indicate that the effect of chemicals incorporated in growth medium is a major factor in distinguishing transcriptomic variance. By applying strict statistical thresholds—a false discovery rate (FDR) of 0.01 and a log2 fold change of > 2 or < −2—we found that 747 and 770 genes were differentially expressed in cells treated with phenol (Figure 3A,C, Appendix A) and formalin (Figure 3B,C, Appendix A) in the non-treated contrast, respectively.

### 3.3. Functional Enrichment Analysis with DEGs

The contrast of phenol and non-treated samples showed 747 genes with differences in their expression. Particularly, 233 and 514 genes were up- and downregulated in the presence of phenol, respectively (Figure 3C, Appendix A). Next, we performed a functional enrichment analysis to identify the classes of DEGs that were overrepresented in a set of *E. piscicida* C07–087 encoded genes. Since 99.4% of the DNA sequence from *E. piscicida* CK108 is identical to a representative *E. piscicida* strain, C07–087, we used the genome information from *E. piscicida* C07–087 [29]. Seven functional categories (C, E, F, G, N, P and T) were considered to be significantly different (FDR less than 0.05) from the DEGs (Figure 4A). Whereas the genes related to cell motility were not found to be upregulated, the downregulated genes were classified into all ranges of the seven significant functional categories. Many differently up- and downregulated DEGs were found in the categories of cell motility, carbohydrate transport and metabolism, nucleotide transport and metabolism, and energy production and conversion (Figure 4A).

Among the 770 DEGs identified from the contrast of formalin and non-treated samples, 261 and 509 genes were up- and downregulated, respectively (Figure 3C, Appendix A). Through the functional enrichment analysis, five significant (FDR less than 0.05) categories (C, E, G, N and P) were identified from the DEGs (Figure 4B). All categories showed that the number of downregulated genes was 2- to 4-fold higher than that of the upregulated DEGs. Interestingly, as found in the phenol functional enrichment, none of the upregulated DEGs were associated with cell motility. These data suggest that the expression of genes (i.e., flagella biosynthetic genes) involved in cell motility is reduced in the presence of phenol and formalin.

### 3.4. Pathway Analysis

To gain knowledge about how metabolic pathways respond to phenol and formalin, pathway enrichment was performed with DEGs using the Kyoto Encyclopedia of Genes and Genomes (KEGG) database. When cells were exposed to phenol, eight pathways were identified based on the cutoff (*p* < 0.05) (Figure 5A). Pathways containing upregulated DEGs were partial two-component systems (EnvZ-OmpR and BaeS-BaeR), histidine biosynthesis and partial ABC transporters (phosphates and phospholipids). On the other hand, pathways harboring downregulated DEGs were the TCA cycle, oxidative phosphorylation, galactose catabolism, partial two component systems (UhpB-UhpC, AtoS-AtoC and PgtC-PgtB), partial ABC transporters (molybdate, iron, nickel, maltose, glutamate/aspartate and oligopeptide), chemotaxis and flagella biosynthesis (Figure 5A).

In total, five pathways were significantly (*p* < 0.05) affected by formalin (Figure 5B). The downregulated DEGs were involved in almost all pathways: the TCA cycle, ABC transporters (molybdate, iron, nickel, glutamine, arginine, glutamate/aspartate and oligopeptide), chemotaxis and flagella biosynthesis. In addition, upregulated DEGs were found in the type 3 and type 6 secretion systems (Figure 5B). Interestingly, the DEGs involved in type 3 and type 6 secretion system were downregulated when phenol was treated (Figure 3C). This suggests that effector molecules are exported to the extracellular environment and can cause host pathogenesis in a specific chemical-dependent manner. Taken together, the pathway analysis results indicate that many metabolic pathways are influenced by phenol and formalin. This experiment also provides evidence that the sub-ICs of chemicals, although the concentrations are not lethal, can act as environmental signaling molecules for aquatic microbial ecosystems.

### 3.5. Validation of RNA-seq Data by qRT-PCR

In order to validate the RNA-seq results, the expression levels of DEGs were further evaluated by qRT-PCR. The expression level of *gapdh* (ETAC_06975), which is widely used in *E. tarda* as an internal control gene [30], was independent from the sub-ICs of phenol and formalin (Appendix A ). Candidate genes (*baeR*, *hisA*, *hcp*, *eseC* and *fliC*) for the validation were chosen from DEGs involved in two-component systems, histidine biosynthesis and motility and secretion systems that displayed increased or decreased transcription levels in response to chemicals (Figure 5, Appendix A). Figure 6 shows that the qRT-PCR results of all representative genes were in good agreement with those of RNA-seq data, presenting the evidence that the genome-wide transcriptomic data in this study provide valuable information that could help to uncover the global mechanism of how *E. piscicida* CK108 responds to chemical exposure.

### 3.6. Motility Analysis

The results of the transcriptome data analysis by DEG identification and functional and pathway enrichment (Figure 3, Figure 4, and Figure 5) suggest that the genes involved in flagella biosynthesis are downregulated in the presence of phenol and formalin. We tested this hypothesis with motility assays. An equal amount of cell culture was spotted in the middle of the soft agar plates which contain the sub-ICs of phenol or formalin and the swimming area was measured after 17 h of incubation. Compared with the non-treated cells, cell movement was dramatically decreased when chemicals were present (Figure 7A). Whereas half of the cell motility was reduced in the presence of formalin, the cells grown in the presence of phenol hardly moved from the inoculated area (Figure 7B). This result shows that treatment with phenol and formalin, even at sub-ICs, causes a significant defect in cell motility. Furthermore, the result of high-throughput transcription analysis showed that DEGs involved in cell motility are downregulated, which corresponds with the phenotypic characteristic that cell motility is reduced when phenol and formalin are present.

## 4. Discussion

Aquatic microbial ecosystems and wastewater treatment systems are often damaged due to microbial growth inhibition caused by the toxicity of aquatic environmental pollutants. In this study, we investigated the impact of the sub-ICs of aquatic pollutants on bacterial gene expression with a marine pathogen *E. piscicida*. We hypothesized that pollutants at the sub-IC would act as environmental signaling molecules and alter the gene expression of the fish pathogen. The strain used in this study was *E. piscicida* CK108, a derivative from CK41 which was isolated from a diseased flounder in the Republic of Korea [20,29]. *E. piscicida* CK41 has approximately 70 kb of the plasmid pCK41. This plasmid encodes genes for virulence factors and resistance to three antibiotics such as kanamycin, tetracycline and streptomycin [20]. We do not rule out a possibility that the plasmid pCK41 can influence the gene expression under chemical exposure, however, there are reasons why *E. piscicida* CK108 was used in this study. First, there was no statistically significant difference in the in vivo virulence assay (LD_50_) between *E. piscicida* CK41 and CK108 with the goldfish and zebra fish [20]. Second, there was also no statistically significant difference at the transcriptional level between the two strains by a pilot RNA-seq experiment. Through these reasons, we used the CK108 strain to perform the transcription analysis on phenol and formalin in this study.

If a pathogen is able to change its pathogenicity by attenuated (avirulent) or hypervirulent factors, we can infer two different scenarios. First, if the pathogen becomes attenuated (or avirulent) by downregulating the gene expression of the virulence factors, it is possible that a bacterium could be camouflaged as normal flora in the host, allowing it to escape the host’s immune system under the existing pollutant conditions. An attenuated (or avirulent) bacterium may easily colonize in a host without interruption of the host immune system. However, once the pollutants are diluted or absent from an aquatic environment, the pathogen in the host restores the gene expression of virulence factors and damages to the host. The second scenario is that a bacterium becomes hypervirulent by the upregulated gene expression of virulence factors. In this case, the pathogen possibly overexpresses its virulence factors (e.g., toxins, biofilm, flagella, fimbriae, etc.), which promotes its ability to carry out adhesion, invasion and penetration on the host cells, increasing its survivability in the host immune cells, which leads to the destruction of hosts even at a low dosage. This study provides the effect of phenol and formalin on a fish pathogen *E. piscicida* CK108. In both scenarios mentioned above, this fish pathogen can cause damages to the fishery industry through environmental pollutants, which induces economic loss.

In our study, with exposure to the sub-IC of phenol and formalin, the major virulence factors of *E. piscicida* CK108, such as the type III secretion system (T3SS), type VI secretion system (T6SS), fimbriae and flagellar/chemotaxis-related genes, altered their expression levels (Figure 5, Appendix A). T3SS and T6SS were identified as the two most important secretion systems in *E. piscicida* [31]. T3SS is a contact-dependent, flagellar-like needle structure that directly injects effectors into eukaryotic host cells using a one-step mechanism. For *Salmonella*, T3SS is important for bacterial internalization and intracellular survival, and is crucial for bacterial infection in mice [32,33]. T3SS is composed of an injectisome (EseB, EseD), effectors (EseG/J/K/H/N/L/M, Trxl), regulators (EsrA-EseE, EsrC) and chaperones (EseC, EscA, EseE) in *E. piscicida* [13]. The effector proteins of T3SS promote NLRC4 and NLRP3 inflammasomes, bacterial colonization and microtubule destabilization [34,35]. In our DEG analysis, T3SS-encoded genes (ETAC_RS04205, 04210, 04215, 04220, 04225, 04230, 04235, 08110) were downregulated with the sub-IC of phenol (Figure 5A, Appendix A), while those genes were upregulated with the sub-IC of formalin (Figure 5B, Appendix A). T6SS is also a contact-dependent, phage tail-like infection apparatus used to puncture and deliver effectors directly into adjacent host or bacterial cells [36]. The spike component called VgrG trimer is responsible for puncturing targeted cells, supported and assisted by the HCP tube [13]. The T6SS effector EvpP prevents NLRP3 inflammasome activation [37]. A previous study found a significant decrease in the transcription level of IL-1β in zebrafish kidney infected with the T3SS mutant and a drastic increase in the transcription level of TNF-α infected with the T6SS mutant when compared with the wild-type [31]. Similar to the T3SS-encoded genes, T6SS-encoded genes (ETAC_RS11445, 11450, 11455, 11460, 11465, 11470, 11475, 11480, 11485, 11490, 11495, 11500, 11505) were downregulated with the sub-IC of phenol (Figure 5A, Appendix A), while diametrical results were detected with the sub-IC of formalin (Figure 5B, Appendix A). These diametrical results of T3SS and T6SS regulation by phenol and formalin suggest that the pathogenesis mechanisms are dependent on the specific aquatic pollutants.

Several two-component systems also modulated the gene expression level under the sub-IC of phenol and formalin. Especially, the two-component systems EnvZ-OmpR and BaeSR were upregulated with the sub-IC of phenol. The EnvZ-OmpR two-component system regulates bacterial virulence by sensing osmolality [14]. This represents a well-known regulatory system that mediates motility, virulence and drug efflux systems in other pathogens [38,39,40,41,42,43]. Moreover, several different efflux systems and regulatory systems were modulated by the sub-IC of phenol (Figure 5A, Appendix A). In *E. coli*, the UhpBC two-component system was modulated in the hexose phosphate transporter, and the AtoSC system was modulated in short-fatty acid chain metabolism [44,45]. With the sub-IC of phenol in *E. piscicida* CK108, the UhpBC two-component regulatory system was modulated, resulting in the PstABCS phosphate transporter system being upregulated and the galactose and maltose transporter systems being downregulated (Figure 5A, Appendix A). The AtoSC fatty acid metabolism system was also modulated, resulting in the MlaBCED phospholipid transporter being upregulated. The BaeSR two component system, which is known to be a regulatory system of AdeABC, AdeIJK, MacAB-TolC and AcrD drug efflux systems in other bacteria, was upregulated [46]. The Afu ABC iron-binding transporter, which can be directly involved in bacterial pathogenesis by modulating iron acquisition, was downregulated [47,48,49,50]. With the sub-IC of formalin, EsrC was upregulated (Appendix A), which is a positive regulator and used to regulate genes encoding the apparatus and effector proteins of T3SS (EseB and EseD) and T6SS (HCP) [51]. As we mentioned above, T3SS and T6SS were upregulated with formalin, which may be a consequence of the upregulation of the EsrC regulator. Not only EsrC but also other regulators were modulated by T3SS and T6SS in *Edwardsiella* sp. The EsrA-EsrB two-component system regulated the injection apparatus proteins, EseB, EseC and EseD, of T3SS [14]. YebC regulates quorum sensing and activates T3SS expression by directly binding to the promoter region of the T3SS gene involved in bacteria colonization in fish [52]. The ferric uptake regulator (Fur) is intertwined with PhoB-PhoR to regulate EsrC expression, and thus to control T3SS/T6SS expression in *E. piscicida* [14]. The PhoQ-PhoP two-component system senses host body temperature, low Mg^2+^ concentration and the presence of antimicrobial peptides, activating T3SS and T6SS [53]. The QseC-QseB two-component system regulates the expression of genes encoding T3SS components by sensing host epinephrine/norepinephrine [54]. For direct regulation of *E. piscicida* T3SS and T6SS, we only detected the regulator EsrC that was upregulated with the sub-IC of formalin, and showed no transcriptional change with the sub-IC of phenol. However, it is possible that phenol and formalin led to a cascade of downstream signaling events without a transcriptional level change of regulators, resulting in virulence factors including T3SS and T6SS being altered in gene expression. Bacterial adhesion to host cells is mediated by fimbriae through the biofilm formation [55,56,57,58,59,60]. The *E. piscicida* type I fimbriae mutant displayed significantly reduced bacterial adhesion to EPC cells, and further, the major fimbrial protein of *E. tarda* possessed vaccine and adjuvant potentials, which means this fimbrial protein is able to induce a specific immune response in the host [61,62]. In our study, both type I and type IV fimbriae genes were downregulated with the sub-ICs of phenol and formalin (Appendix A). Bacterial motility is also critical to infect the host [63,64,65,66]. The flagellum of a bacterial pathogen is associated with its infectious life cycle, as it allows escape from the host immune cell attack and cellular invasion to the host cells [67]. In addition, the flagellin protein is a major bacterial antigen, H antigen, which could be targeted in host immune systems and acts as an immune adjuvant molecule [68,69]. The interaction between the bacterial flagellin protein and toll-like receptor 5 could trigger the pro-inflammatory response in hosts [70]. Bacterial flagella also contribute to biofilm formation and are correlated with T3SS. Several bacteria exhibit attenuated infectivity when they have lost motility or flagella/chemotaxis-related genes. In our study, flagella/chemotaxis-related genes were downregulated (Figure 5, Appendix A), and as well, the motility phenotype dramatically reduced with the sub-ICs of phenol and formalin (Figure 7). Collectively, several different regulatory systems were modulated by the sub-ICs of phenol and formalin in *E. piscicida* CK108, which modulated motility, virulence, drug and ion efflux, sugar and amino acid transporters and energy metabolism-related gene expression levels. Both phenol and formalin influenced genes related to energy production (i.e., the TCA cycle), amino acid and carbohydrate transporters and cell motility. Reduced motility on semi-solid medium with the chemicals supported the transcriptome results. Moreover, upregulation of the histidine pathway and secretion system were specific to the growth conditions of phenol and formalin, respectively. These enormous gene modulations in bacteria by sub-IC environmental pollutants could modulate not only the growth of cells but also their virulence, pathogenesis, drug resistance, stress response, energy metabolism and so on.

Results in this study provide a new aspect of environmental pollutants as signaling molecules that modulate gene expression in pathogenic bacteria. With exposure to the sub-ICs of phenol and formalin, the major virulence factors of the *E. piscicida* CK108 strain were up- or downregulated, which may alter the pathogenesis mechanisms of pathogens in the hosts. Although the major metabolic pathway (the TCA cycle) was inhibited, the production of essential amino acids (e.g., histidine) was enhanced with phenol. Motility, chemotaxis and biofilm-mediated fimbrial expression were downregulated. T3SS and T6SS were upregulated with formalin. The results of this study demonstrate the comprehensive molecular response of microorganisms to environmental chemicals. Additionally, we provided scientific evidence of the importance of the bioremediation of environmental contaminants, even when discharged at low concentrations, and contribute potential knowledge about pathogen control in the fishery industry.

## Figures and Tables

**Figure 1 microorganisms-08-01068-f001:**
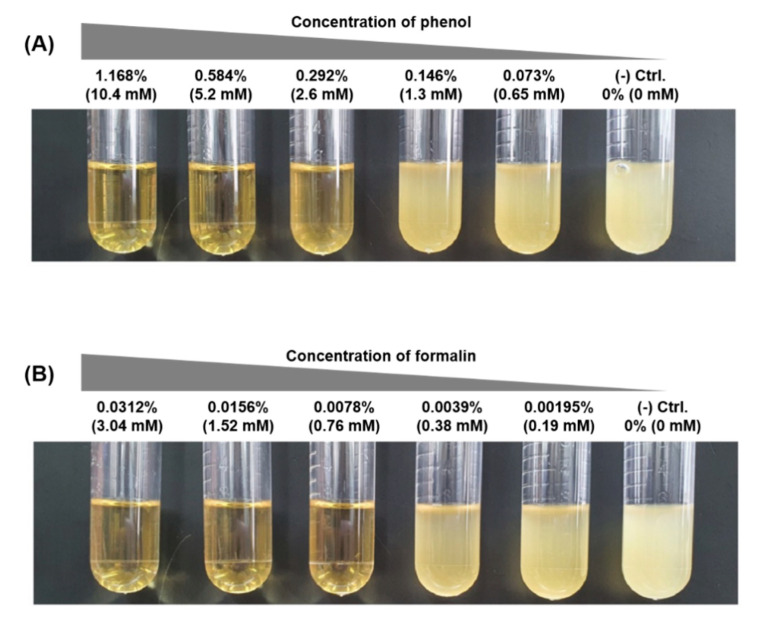
Identification of the sub-inhibitory concentration (sub-IC) for (**A**) phenol and (**B**) formalin. Cells were grown in 3 mL of Brain Heart Infusion (BHI) medium with the indicated percentages of phenol and formalin, respectively. The image was taken after 24 h of incubation at 27 °C.

**Figure 2 microorganisms-08-01068-f002:**
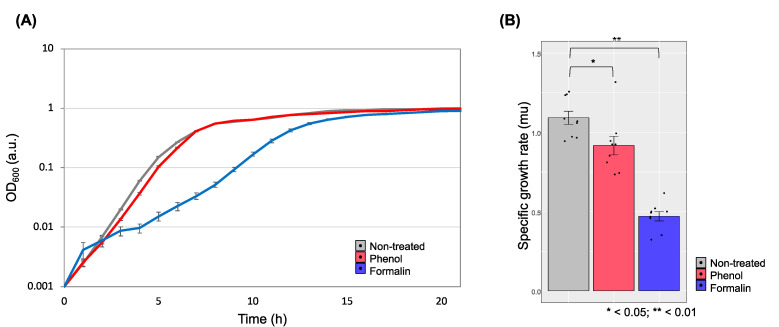
Growth of *E. piscicida* CK108 in BHI broth with the sub-ICs of phenol and formalin. (**A**) Growth monitoring in a 96-well microplate over time. The optical density (OD_600 nm_) is indicated by log-scale. (**B**) Comparison of specific growth rates at the exponential phase. A total of *n* = 9 (biological triplicate and technical triplicate) samples were tested. Error bars represent the standard deviation of the mean. Asterisks represent significant differences between the growth conditions as determined by *t*-tests.

**Figure 3 microorganisms-08-01068-f003:**
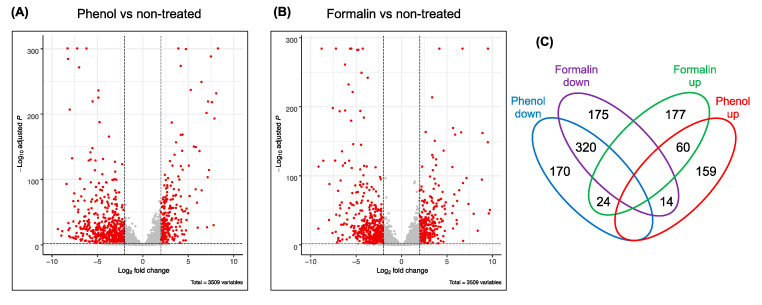
Analysis of RNA-seq data. (**A**) Determination of the differentially expressed genes (DEGs) from the contrast of phenol and non-treated samples (see also Appendix A). Each gene analyzed can be visualized as a dot in the volcano plot. Red dots indicate genes meeting the cutoff (log2 change, 4-fold; false discovery rate < 0.01). (**B**) Comparison of DEGs from formalin and non-treated samples (Appendix A). (**C**) Venn diagram indicating the number of DEGs across the comparisons to non-treated samples and the overlap between each set of genes.

**Figure 4 microorganisms-08-01068-f004:**
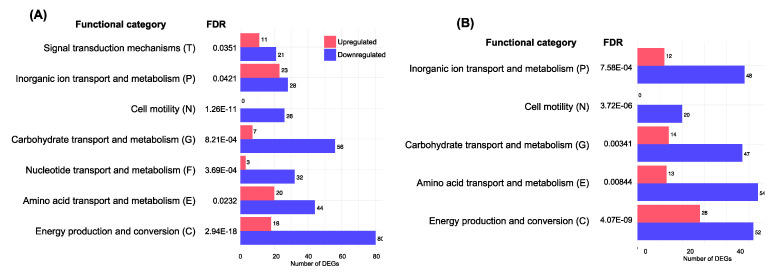
Functional enrichment analysis with DEGs. (**A**) Comparison of DEGs from phenol and non-treated samples. (**B**) Comparison of DEGs from formalin and non-treated samples. Numbers next to the horizontal bar indicate the number of DEGs analyzed.

**Figure 5 microorganisms-08-01068-f005:**
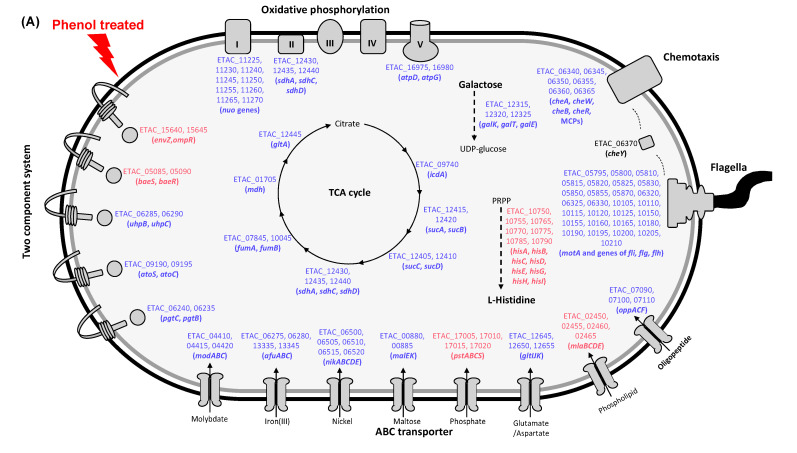
Pathway enrichment analysis based on the Kyoto Encyclopedia of Genes and Genomes (KEGG) database. (**A**) Comparison of DEGs from phenol and non-treated samples. (**B**) Comparison of DEGs from formalin and non-treated samples. Upregulated DEGs and downregulated DEGs are indicated in red and blue, respectively. Proteins and cellular apparatuses are not to scale.

**Figure 6 microorganisms-08-01068-f006:**
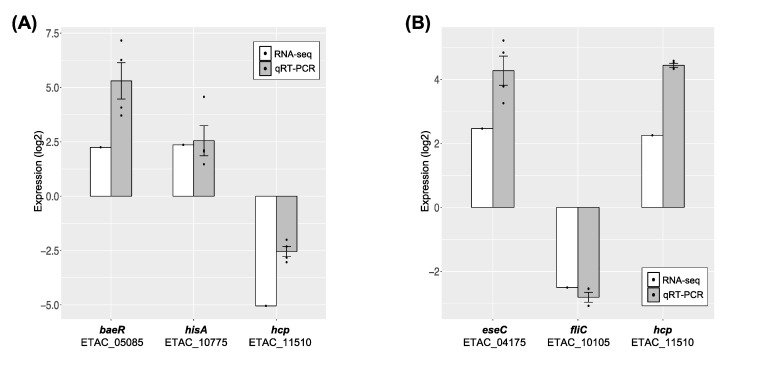
Validation of RNA-seq data by qRT-PCR. The expression levels of the selected genes in qRT-PCR were quantified through a comparison with the internal control gene of *gapdh* when cells were grown in the sub-ICs of (**A**) phenol or (**B**) formalin. A total of *n* = 4 (biological quadruplicate) samples were applied. Error bars represent the standard deviation of the mean.

**Figure 7 microorganisms-08-01068-f007:**
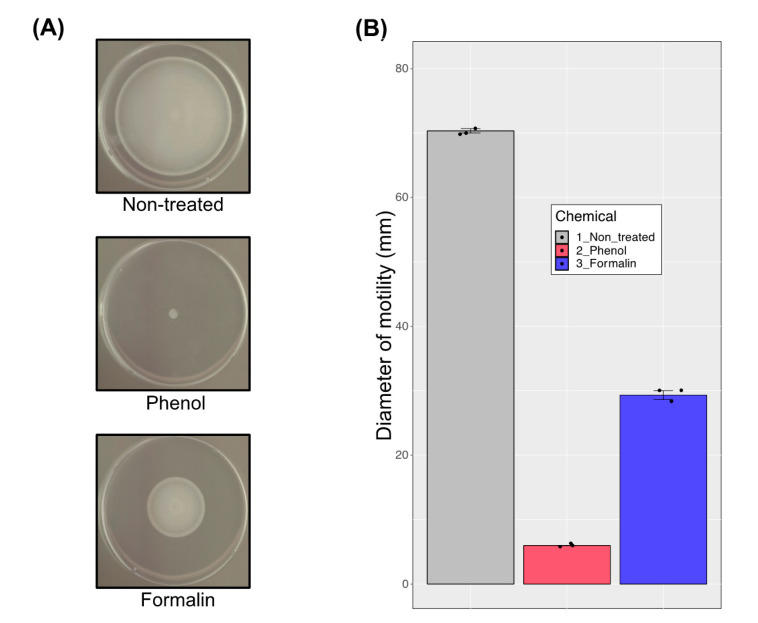
Movement of *E. piscicida* CK108 on solid BHI medium with the sub-ICs of phenol and formalin. (**A**) Representative cell motility from three independent assays. (**B**) Measurement of the swimming area. A total of *n* = 3 (biological triplicate) samples were applied. Error bars represent the standard deviation of the mean.

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
