# Peer review of "In vitro Edwardsiella piscicida CK108 Transcriptome Profiles with Subinhibitory Concentrations of Phenol and Formalin Reveal New Insights into Bacterial Pathogenesis Mechanisms"

_microorganisms, 2020, doi:10.3390/microorganisms8071068_

Round 1
Reviewer 1 Report
This is a very solid study on Edwardsiella piscicida response to Subinhibitory Concentrations of Phenol and Formalin. Although the implication of the findings on pathogenesis in vivo is unclear, the data are valuable to the field.
Comments:
- how many biological replicates are included in the two RNA-seq studies? Please specify.
- Figure 3C. The comparison of Phenol and Formalin data sets could be improved. For example, there should be one panel showing "phenol-upregulated vs. formalin-upregulated", another panel showing "phenol-downregulated vs. formalin-downregulated".
- Any guess on the potential mechanisms for the upregulation of secretion systems? Suggest adding a paragraph to the discussion section.
Author Response
Please see the attachment. Thank you for the great comments.

Reviewer 2 Report
The proposed manuscript to Microorganisms journal, entitled “Edwardsiella piscicida Transcriptome Profiles with Subinhibitory Concentrations of Phenol and Formalin Reveal New Insights into Bacterial Pathogenesis Mechanisms”, reports the results of the in vitro action of formalin and phenol on the gene expression of Edwardsiella piscicida CK108. Formalin and Phenol were used at sub-inhibitory concentrations.
The article by Ju Bin Yoon, Sungmin Hwang (co-first authors) et al. is interesting although need to be well reviewed. Experimental data appear to be well thought out but at the same time they are poorly or not well discussed.
Minor revision
Add in the title the name of the used strain, CK108. Results are pertinent with this strain and not, in general, with the species. Different strains may generate different results. Further the assays were made in vitro and not in vivo. This should be indicate.
Abstract
It seems OK although need to report specific detected relevant gene.
Introduction
Line 72
Revise the following phrase: “..when E. piscicida CK108, a large antibiotic-resistance plasmid-cured strain of CK41, is 72 exposed..”. The connection between the two strains is not clear.
Line 74
Remove the phrase: “From the biotechnology point of view…”. The article does not contain biotechnological information.
Results
Sec. 3.1. and in general.
Specify the in vitro assays. The authors does not made the in vivo assays, using RNA directly extracted from the sea water prokaryotic sample.
Major revision
Introduction
It is quite poor and at times vague. Little information on pathogenicity mechanisms.
The pathogenicity mechanisms are an integral (focus) part of the title, so these should be introduced with particular attention. No gene (and related references) is described. For instance, why the author do not introduce info on the Edwardsiella piscicida pathogenic mechanisms, involved genes and what is known. Introduce article describing the effect of pollutants, under sub-inhibitory concentrations, on bacterial cells in general. Are there studies describing the in vivo or in vitro effect ?
The authors introduce such phrase “..a large antibiotic-resistance plasmid-cured strain of CK41”. What does it means ?!!. Describe, briefly, such plasmid and its own characteristics! And its transcripts ??!!
No information on the bio-molecular action of formalin and phenol.
Finally, it is very important to specify that the experimental data are in vitro detected. We don’t know what happen in vivo. The different chemical-physical conditions, the various biotic and abiotic elements present in vivo can generate different results.
Results
Sec. 3.3.
Is too generic/vague. Describe in more detail. Type III secretion system (T3SS), type VI secretion system (T6SS), etc, are mentioned in the discussion but in the results they do not appear.
What are the exclusive genes activated by formalin? The Venn diagram (fig 3C) tell as 352, but no mention of these is in the text, at least the most important from a pathogenic point of view. Idem for the exclusive 329 genes regulated by phenol, as well as the common genes. Why formalin has a major impact on E. piscicida CK108 growth ?! Among the 352 genes are there some more incisive ? Are genes that regulate the transcription of pathogenic pathways? This result is interesting and need to be addressed.
Discussion
It is not incisive, well defined. It is also too short. This appears when the authors have no clear idea of their results. The authors have results regarding the interference of formalin and phenol on the gene expression of Edwardsiella piscicida CK108, using sub-inhibitory concentrations. What emerges is that formalin has more inhibiting capacity than phenol. This result is not well discussed. Why formalin is more stronger that phenol. Among the 352 gene controlled by the formalin alone there are some encoding for important pathogenic factors, transcription factors and so on.
What are the differences and meaning between the up- and down-regulated unshared genes ?
Among the down-regulated genes, are there encoding for transcription factors acting on some genes up-regulated ? Are there functional correlations ?
Further, the results regarding (just for instance) the type III secretion system (T3SS) and type VI secretion system (T6SS) should be well discussed. With the phenol they are up-regulated but with the formalin are down-regulated. What it means ? The answer of the authors into the discussion is vague. Most of the discussion (and results) describe generic functional category: Energy, signal, carbohydrate, etc. In my opinion among the genes up- or down-regulated detected for each category should be made a more painstaking study. An alteration of the most important metabolic pathways, attributable to amino acids, carbohydrates, nucleic acids, transport etc is more or less obvious, when a bacterium is subjected to stress.
The unique well definite results is the cell motility. The phenol has a more effective action than formalin (fig. 7A and B). Anyway the motility and the grow of Edwardsiella piscicida CK108 seem to be independent. Phenol has a minor action than formalin (fig. 2B).
Finally, the author should well revise the discussion. They should deepen the study of their results to better assemble the discussion.
Author Response

(The authors gave the same response as above.)

Reviewer 3 Report
The article” Edwardsiella piscicida Transcriptome Profiles with Subinhibitory Concentrations of Phenol and Formalin Reveal New Insights into Bacterial Pathogenesis Mechanisms“ provide interesting information. However, I believe there is lack of novelty for this research. Here, I have added few comments that may help authors for improving the manuscript.
Line 39 , after “microorganisms that thrive in aquatic habitats.” I suggest to use some references that contribute to the presence of phenolic compounds and their effects on the aquatic organisms such as https://www.sciencedirect.com/science/article/pii/S2666188820300101
Or https://www.ingentaconnect.com/content/ben/mroc/2017/00000014/00000005/art00006
Line 46: reference[2] is not very strong, phenolic compounds are readily biodegradable and I donot belive they persistent long in the environment. You may focus on them as residual of UV filters and many other pharmaceuticals or as biotransformation products
Line 54 to 55, authors suddenly jumped to “Edwardsiella piscicida “
Line 71” use the past tense “used” instead of use
Line 80, Reference [15] is not at appropriate place to reference it.
Line 85: what is the applied wavelength
Line 91:How did you prepare the phenol or formalin solutions? It was used as it was purchased?
Line 92—103, the method miss the appropriate references to support their method
I suggest to put the Figure 1 to SI document
Pleas provide a section in material method for the statistical analysis
Author Response

(The authors gave the same response as above.)

Round 2
Reviewer 2 Report
Dear Authors,
All revisions were made with precision and attention. The revised version of the manuscript is now more incisive and therefore publishable in Microorganisms. The authors, experimentally, did a good work.